# Improved Dimensional Stability and Mold Resistance of Bamboo via In Situ Growth of Poly(Hydroxyethyl Methacrylate-*N*-Isopropyl Acrylamide)

**DOI:** 10.3390/polym12071584

**Published:** 2020-07-16

**Authors:** Tingsong Liu, Wenhao Zhang, Jie Wang, Yan Zhang, Hui Wang, Fangli Sun, Lili Cai

**Affiliations:** 1School of Engineering, Zhejiang A & F University, Hangzhou 311300, China; Liutingsong94@outlook.com (T.L.); as291884712@163.com (W.Z.); wj1843520@163.com (J.W.); zhangy@iccas.ac.cn (Y.Z.); wanghui@zafu.edu.cn (H.W.); 2Department of Forest, Rangeland and Fire Sciences, University of Idaho, Moscow, ID 83844, USA

**Keywords:** bamboo, hydroxyethyl methacrylate, *N*-isopropyl acrylamide, in situ polymerization, dimensional stability, mold resistance

## Abstract

Bamboo is a natural and renewable building material but its application has been limited due to the low dimensional stability and poor durability against mold. In this study, monomers of hydroxyethyl methacrylate (HEMA) and *N*-isopropyl acrylamide (NIPAM) were impregnated in bamboo to facilitate the in situ growth of poly-HEMA and NIPAM (PHN) copolymer. Prior to that, the effects of different reaction conditions, including the molar ratio of HEMA to NIPAM and their concentrations, the amount of initiator (ammonium persulfate, APS) and crosslinking agents (*N*,*N*′-Methylenebisacrylamide (MBA), and glutaric dialdehyde (GA)) on the swelling capacity of PHN were optimized. The formation of PHN was confirmed by using Fourier transform infrared spectroscopy and thermogravimetric analysis, which shows the characteristics peaks of both HEMA and NIPAM, and increased pyrolysis and glass transition temperatures, respectively. After impregnation of PHN pre-polymerization formulation to bamboo, it was observed that PHN filled most of the pits in the bamboo cell wall and formed a tight network. Moreover, the dimensional stability of PHN treated bamboo was significantly improved with an anti-swelling efficiency of 49.4% and 41.7%, respectively, after wetting–drying and soaking–drying cycles. A mold infection rate of 13.5% was observed in PHN-treated bamboo as compared to a 100% infected control group after a 30-day mold resistance test. Combined results indicate that in situ polymerization of HEMA and NIPAM in bamboo is a promising method to develop exterior used bamboo products with enhanced dimensional stability and mold resistance.

## 1. Introduction

As one of the fastest-growing plants in the world, bamboo offers a natural and sustainable alternative for various applications such as flooring, decking, wall-covering, etc. [1]. However, bamboo is susceptible to dimensional changes as it only consists of longitudinal fibers [2]. It is also prone to mold and fungi attack because of the relatively high moisture content in initial processing stages and accumulated sugar and starch in culm [3].

A lot of efforts have been made to address these issues [4,5]. In situ polymerization has been demonstrated as one of the most effective ways to improve the dimensional stability and mold resistance of bamboo. Examples include impregnating bamboo with different treating solution, such as furfuryl alcohol, styrene, and polyester, followed by heat treatment to induce the polymerization reaction [6,7,8]. The occupation of these modifying agents in the cell cavity or the cell wall leads to less space for water molecules, thus improve the dimensional stability and resistance against mold fungi [9]. However, the performance of bamboo after these treatments is still far behind the desired quality. For example, the elastic modulus of furfuryl alcohol modified bamboo decreased and the intrinsic warm color of bamboo turned into dark brown [10]. The high self-polymerization tendency of styrene and low reactivity toward non-conjugated double bonds have limited its application in bamboo modification [11]. There are other ways to modify bamboo, such as using monomers, which have significantly enhanced the properties of bamboo [12]. However, few of them have been able to introduce the polymer at the cell wall level.

Hydroxyethyl methacrylate (HEMA) and *N*-Isopropyl acrylamide (NIPAM) are two readily commercially available monomers with very low molecular weights. Both HEMA and NIPAM are cytophobic, biocompatible, chemically and thermally stable, and therefore, have been extensively used in various biomedical applications [13,14]. HEMA and NIPAM monomers can form copolymer by a simple and mild synthesis route, which considerably reduces energy consumption and improves economic efficiency. This process typically uses ammonium persulfate (APS) as an initiator with a polymerization temperature between 60 and 100 °C [15]. In addition, the amide group in NIPAM may form hydrogen bonds with the hydroxyl group in HEMA or cellulose [16]. Thus, it is possible to obtain bamboo products with improved dimensional stability and mold resistance by introducing these two monomers and facilitating the in situ growth of Poly(HEMA–NIPAM) (PHN) in bamboo. Moreover, the application of HEMA–NIPAM copolymer for bamboo treatment has not been investigated.

In this study, we first optimized the conditions to synthesize PHN polymer with the lowest swelling capacity and then treated bamboo with the optimized formulations. The distribution of PHN in bamboo was visualized using SEM and quantified by volume percentage gain and weight percentage gain. The dimensional stability and mold resistance of the treated bamboo were also evaluated.

## 2. Materials and Methods

### 2.1. Materials

Monomers hydroxyethyl methacrylate (HEMA, 96%) and *N*-Isopropyl acrylamide (NIPAM, 98%) were purchased from Macklin Co., Ltd. (Shanghai, China), both of which contain a stabilizer of Hydroquinone monomethyl ether (MEHQ). Ammonium persulfate (APS, 98%), *N*,*N*′-Methylenebisacrylamide (MBA, 99%), and Glutaric dialdehyde 25% water solution (GA) were purchased from Aladdin Co., Ltd. (Shanghai, China). Deionized water was used throughout the experiments.

Bamboo specimens (4-year-old *Phyllostachys edulis*) were obtained from a local forest in Hangzhou city, Zhejiang Province, China. The outer epidermis and inner pith rings of the bamboo were planed off. The resulting samples were further processed into sizes of 20 × 20 × 4.5 mm^3^ (*L* × *T* × *R*) and 50 × 20 × 4.5 mm^3^ (*L* × *T* × *R*) for dimensional stability test and mold resistance test, respectively.

### 2.2. Synthesis of PHN Polymer

The main factors affecting the polymerization of PHN include the molar ratio of HEMA to NIPAM and their concentrations, the amount of initiator (APS), crosslinking agents (MBA and GA), as well as the reaction temperature (*t*). Water absorption capacity (*P_w_*) and volume swelling rate (*P_v_*) of the synthesized polymers were used as indicators as a lower *P_w_* and *P_v_* are more desirable for bamboo treatment. Single-factor experiment design was conducted to obtain the polymers with desired properties.

To determine the effects of monomer on *P_w_* and *P_v_*, different molar ratios of HEMA to NIPAM were dissolved in deionized water at designed concentrations, as shown in Table 1. Then the initiator (1 wt.% APS) and crosslinking agents (1 wt.% of both MBA and GA) were added to the above solutions, followed by heat curing at 80 °C for 2 h. The reaction temperature effects were tested using four different temperatures (70, 80, 90, and 100 °C) under the following fixed conditions: reaction time of 2 h, molar ratio of HEMA to NIPAM at 5:1, as well as the amount of APS of 1 wt.%, MBA of 1 wt.%, and GA of 1 wt.%. For initiator effects, different amounts of APS at 0.5, 1.0, 1.5, and 2.0% to the total mass of HEMA and NIPAM were used while the molar ratio of HEMA to NIPAM of 5:1, and both MBA and GA at 1 wt.% were added as crosslinking agents. The reaction lasted for 2 h at 80 °C. For cross-linking agents, three treatments were involved, including 1 wt.% MBA, 1 wt.% GA, and 2 wt.% MBA and GA (at equal mass ratio). One percent of APS was used and the reaction was carried out at 80 °C for 2 h. The calculation of cross-linking agents and the initiator was based on the total mass of HEMA and NIPAM. For solution concentration effects, all the experimental conditions were the same except that a concentration of HEMA and NIPAM (molar ration of 5:1) at 10%, 20%, and 40% were used.

The water absorption capacity (*P_w_*) and volume swelling capacity (*P_v_*) of PHN were tested by exposing the synthesized polymer to two cycles of water immersion and oven dry. For each cycle, the samples were immersed in DI water for 24 h with water exchange at an interval of every 8 h to remove the unreacted monomers, followed by oven-drying at 80 °C to a constant weight. The volume (V, cm^3^) and mass (M, g) of the wetted and oven-dried samples in each cycle were measured using Archimede’s method and a balance, respectively. *P_w_* and *P_v_* of polymers were calculated according to Equations (1) and (2), respectively. All the results were reported as the average of the two tests.
(1)$$P_{w}=\frac{M_{1}-M_{0}}{M_{0}}$$
(2)$$P_{v}=\frac{V_{1}-V_{0}}{V_{0}}$$
where *V_0_* and *M_0_* represent the volume and mass of the samples after oven-drying while *V_1_* and *M_1_* denote as the volume and mass of the samples after 24 h water immersion.

### 2.3. Characterization of PHN Polymer

FTIR spectra of Poly-hydroxyethyl methacrylate (PHEMA), Poly-N-isopropyl acrylamide (PNIPAM), and PHN were collected by using Shimadzu IR Spectrophotometer (IRPrestige-21, Shimadzu, Japan) in a wavenumber range of 400 to 4000 cm^−1^ at a resolution of 4 cm^−1^. The specimens were mixed with KBr at a weight ratio of 1:100 and the mixture was grounded into even and fine powders, followed by compressing at 160 MPa to form a pellet. FTIR spectra of the specimen pellets were recorded at 32 scans per specimen. All the spectra were baseline corrected and normalized before further analysis.

Thermogravimetric analysis (TGA) of PHEMA, PNIPAM, and PHN were performed by using a thermogravimetric analyzer (TG 209 F3 Tarsus, Netzsch, Germany). The specimens were ground into 100 mesh homogeneous particles and weighed to 6 ± 1 mg in an alumina pan. Then the samples were heated from 40 to 600 °C at a rate of 10 K·min^−1^ under N_2_ with a flow rate of 20 mL·min^−1^. Differential scanning calorimetry (DSC) analysis was carried out using a DSC Q2000 instrument (TA Instruments, New Castle, DE, USA). Two steps were included when conducting the test. The first step includes heating the specimens from room temperature to 200 °C at 10 K·min^−1^ and subsequently cooling to −40 °C at 10 K·min^−1^. For the second step, the specimens were reheated from −40 to 220 °C at a heating rate of 5 K·min^−1^.

### 2.4. Preparation of PHN Treated Bamboo through In Situ Growth

The formulation of the synthesized PHN with the lowest volume swelling capacity (*P_v_*) was further used to treat bamboo samples and the treating solution was prepared as follow: HEMA and NIPAM were dissolved at a molar ratio of 5:1 in DI water, followed by adding 1 wt.% of initiator and 2 wt.% of crosslinking agents (equal weight ratio of MBA and GA) under constant stirring. Three concentration gradients of the above solutions were used to treat bamboo, as shown in Table 2. Positive control includes untreated groups while negative controls contain HEMA and NIPAM treated bamboo at corresponding concentrations.

The dried specimens were impregnated with the above treating solutions under a vacuum of 0.09 MPa for 20 min, followed with a pressure of 0.7 MPa for 30 min. The specimens were removed from the treating tank and the residual solution on the surface of the specimens was gently wiped off. Finally, the wet specimens were wrapped in tinfoil and further heated at 80 °C for 2 h to promote the in situ polymerization.

### 2.5. Scanning Electron Microscope (SEM) Observation of PHN Polymer in Bamboo

Surface morphologies of bamboo and polymer slice surface were observed using a scanning electron microscope (SEM, TM3030, HITACHI, Tokyo, Japan). Before SEM observation, bamboo specimens, both in cross-section and along the longitudinal direction, were sliced into 30 μm by a microtome. The polymers were freeze-dried and the fault surface was obtained by the freeze-fracture method. These specimens were fixed to aluminum stubs using carbon adhesive tapes, followed by sputter-coating with gold at a thickness of 10 nm under vacuum before the SEM examination. The binarization images were obtained using Matlab (R2017a, MathWorks, Natick, MA, USA) while the cell wall thickness of bamboo was measured using Nano Measurer (V1.2, Fudan University, Shanghai, China).

### 2.6. Weight Gain and Volume Gain

The retentions of polymer in bamboo were indicated by volume percent gain (VPG, %) and weight percent gain (WPG, %), which were calculated according to Equations (3) and (4), respectively. The length, width and height of specimens were measured by a caliper while the weight is determined by an electronic balance.
(3)$$\mathrm{VPG}=\frac{v_{t}-v_{0}}{v_{u}}$$
(4)$$\mathrm{WPG}=\frac{m_{t}-m_{0}}{m_{0}}$$
where *v_0_* and *v_t_* are the volume of bamboo specimens before and after treatment, cm^3^, respectively. *m_0_* and *m_t_* are the weight of specimens before and after treatment, g, respectively.

### 2.7. Dimensional Stability Test

To evaluate the dimensional stability of modified bamboo, the specimens were exposed under simulated weathering conditions either consisting of three drying-soaking cycles or three drying-wetting cycles. Wetting here means exposing the samples at a relative humidity of 85 ± 5% and temperature of 20 ± 2 °C for 3 days while soaking means the samples were immersed in water at 25 ± 2 °C for 3 days. The samples in both weathering conditions were oven-dried at 80 °C for 12 h. The volumes of bamboo samples at each drying and wetting/soaking cycle were recorded as *V_dn_* and *V_wn_*, respectively.

The volumetric swelling coefficient (*S_w_*) and anti-swelling efficiency (ASE) were calculated according to Equations (5) and (6), respectively. Only the ASE of the third cycle is discussed in this paper.
(5)$$S_{w}=\frac{\left(V_{wn}-V_{dn}\right)}{V_{dn}}\times 100$$
(6)$$\mathrm{ASE}=\frac{\left(S_{w0}-S_{wk}\right)}{S_{wk}}\times 100$$
where *V_wn_* and *V_dn_* is the volume of the specimens after the *n*th wetting/soaking and drying, respectively. *S_w0_* and *S_wk_* represents the volumetric change rate of the untreated specimens and treated specimens, respectively.

### 2.8. Mold Resistance Test

Mold resistance of the treated bamboo was tested following standard AWPA E24-12. The treated bamboo strips (50 mm × 20 mm × 4.5 mm) were placed in the potato dextrose agar (PDA) petri dishes, which were pre-inoculated with a mixture of three types of fungi, namely *Aspergillus niger*, *Trichoderma viride*, and *Penicillium citrinum*. The cultures were further incubated in an environment chamber at a temperature of 25 °C and relative humidity of 85% for 30 days. The average area of the samples infected by mold was recorded every two days over the incubation period.

### 2.9. Statistical Analysis

The data of VPG and WPG were analyzed by Statistical Product and Service Solutions (SPSS V25.0, IBM Corp., Armonk, NY, USA). The results were interpreted at a 95% confidence interval.

## 3. Results Discussion

### 3.1. Effect of Synthesis Conditions on the Swelling Capacity of PHN Polymer

To obtain PHN with a low swelling capacity, single-factor experiments were designed based on the following factors: molar ratio of HEMA to NIPAM and their concentrations, the amount of initiator (APS) and crosslinking agents (MBA and GA) as well reaction temperature (*t*). The swelling behaviors, i.e., water absorption capacity (*P_w_*) and volume swelling capacity (*P_v_*), of all the polymers, are closely related to their structures [17]. Thus, to reduce their swelling capacity, physical or chemical network crosslinking have been commonly adopted and the latter one has been found to be more efficient [18].

The effect of the monomer ratio on swelling capacities of PHN is shown in Figure 1A. As the molar ratio of HEMA to NIPAM decreased, the swelling capacity of polymer slightly decreased and reached a minimum of *P_w_* and *P_v_* at 17.4% and 16.6%, respectively, at a molar ratio HEMA to NIPAM of 5:1. These numbers rose dramatically to 287% and 175% when the molar ratio of HEMA to NIPAM is 1:2. This fact may be related to the strong hygroscopicity of PNIPAM homopolymer produced from the excessive NIPAM [19].

The influence of reaction temperature on swelling capacities of the polymer network is shown in Figure 1B. At a low reaction temperature of 70 °C, *P_v_* and *P_w_* of PHN are relatively high due to the high swelling capacities of incompletely polymerized networks. As reaction temperature increases to 80 °C, both *P_w_* and *P_v_* of PHN are the lowest at 16.8% and 19.4%, respectively. The further increase of reaction temperatures to 90 and 100 °C results in high swelling capacities of PHN again. This phenomenon may be due to the dead-end polymerization or even a flash polymerization of PHN networks, which are induced by the large amounts of free radicals from APS at elevated temperatures [20].

Figure 1C shows the correlation between the initiator amount and the swelling capacities of PHN polymers. The overall swelling capacities of polymer networks increased as a function of initiator amount, with a higher variation in volume swelling capacity than that in weight swelling capacity. This is because during radical polymerization, the higher the initiator amount, the more the active centers will be and the shorter the polymer chain will be formed [21], which results in higher water swelling of the PHN. On the contrary, a lower amount of initiator could lead to a lower initiation efficiency and need a longer time for polymerization. Considering bamboo contains some polymerization inhibitors, such as phenols and tannins, an initiator of 1 wt.% was selected as this formulation provides a relatively low *P_w_* and *P_v_* at 10.5% and 14.6% respectively.

The spatial structures of the final polymer are closely related to the crosslinking agents used during the polymerization process. In this study, two types of crosslinkers, MBA and GA, were chosen and their effects on the swelling capacities of PHN were shown in Figure 1D. When only adding MBA, a three-dimensional spatial structure was postulated to be formed due to the electrophilic addition of the double bonds between each compound. Thus, a lower swelling capacity of PHN was obtained as compared to that of the control. On the other hand, when GA is used as a crosslinking agent alone, it is hypothesized that GA will react with hydroxyl groups in HEMA and with the secondary amine in NIPAM under acidic conditions, which gives a more compact polymer structure with dramatically reduced swelling capacities (Figure 1D,F). Although the addition of both types of crosslinking agents (1 wt.% MBA and 1 wt.% GA) does not contribute to a lower swelling capacity (*P_w_* and *P_v_* at 10.3% and 8.1%, respectively) in comparison to the one only using GA as a cross-linking agent, a more complex and tight polymer network is hypothesized to be formed as MBA and GA reacts with different functional groups on the main chain and form two different and entangled polymer networks [22]. However, the hydrophilic amino group of MBA in the polymer chain could compromise the swelling capacities of PHN, thus not significantly improving the swelling capacity.

The effect of solution concentration on the swelling capacities of PHN network is shown in Figure 1E. The swelling capacities of PHN showed a downward trend as a function of solution concentration, in which the *P_w_* (from 56.6% to 9.8%) decreased more rapidly than *P_v_* (40.0% to 23.9%). This observation is related to the increased amounts of radicals per unit volume at higher concentrations and the increased reaction rates between the monomers and the main chain. These conditions allow for a higher reaction rate and lead to a tighter polymer network [23].

### 3.2. Formation of PHN Polymer

FTIR spectra of PHEMA, PNIPAM and PHN are shown in Figure 2. In the spectrum of PHEMA, the characteristic peaks at 3438 and 1728 cm^−1^ are attributed to –OH and –C=O, respectively. PNIPAM shows a characteristic absorption peak of N–H at 3310 cm^−1^ and at 1548 cm^−1^, which corresponds to its stretching vibrations and deformation vibration, respectively. The peak at 1648 cm^−1^ is related to C=O stretching [20]. Those characteristic peaks in both PHEMA and PNIPAM can be observed on PHN, which indicated the formation of PHN.

The thermal stabilities of PHEMA, PNIPAM, and PHN were studied by using thermogravimetric analysis (TGA). As shown in Figure 3, for all the polymers, the weight loss below 150 °C is related to the loss of water. The initial decomposition temperature (*T_i_*) of PHEMA is 319 °C and its thermogram represents two decomposition stages, with a degradation temperature at the maximum rate (*T_m_*) at 347 and 414 °C in the first and second stage, respectively. In these stages, PHEMA could break down as a monomer of HEMA and at higher temperatures continue to decompose into small molecular compounds, such as diethylene glycol, ethylene glycol, and acetaldehyde [24]. For PNIPAM and PHN, only one decomposition stage was observed. For example, the *T_i_* of PNIPAM was 347 °C, much higher than that of PHEMA and its derivative thermogravimetry (DTG) curve shows a sharp peak with a *T_m_* at 403 °C. This stage could be attributed to the decomposition of the side chain and even the skeleton structure of PNIPAM. In terms of PHN, the weight loss of copolymer takes place in a slightly different way with *T_i_* at 360 °C, which is higher than those with PNIPAM and PHEMA, while the *T_m_* of PHN is 407 °C and the peak was relatively flat. These results indicate that the improved thermal stability of PHN as compared to those of PHEMA and PNIPAM, which could be related to the formation of the copolymer through covalent bonds, as well as the intramolecular and intermolecular hydrogen bonds between –NH and –OH in the copolymer chains.

The DSC curves of PHEMA, PNIPAM, and PHN are shown in Figure 4, all of which have an endothermal baseline shift associated with the glass transition temperature (*T_g_*) at 95, 125, and 118 °C, respectively [25,26]. As shown in the DSC curve of PHN, there is no peaks showing crystallization or melting in the heating range. These results indicate PHN polymer is amorphous with complex sidechains distributed randomly along the main chain.

The morphologies of PHEMA, PNIPAM, and PHN are shown in Figure 5. The microstructures of both PHEMA (Figure 5A) and PNIPAM (Figure 5B) are porous with the later possessing much larger pore sizes than those in the former. In contrast, a fine and dense lamellar structure was observed in PHN at the same magnification (Figure 5C), which could result from the polymerization between PHEMA and PNIPAM. In fact, those fine architectures compromise micro-pores with rugged appearance when observed further at a magnification of 3000× (Figure 5D). The compact microstructure of PHN provides limited access for water to pass through, consequently resulting in lower swelling capacities than those with PHEMA and PNIPAM, which could facilitate better dimensional stability when polymerizing in bamboo.

### 3.3. PHN Polymer in Bamboo

Figure 6 shows the SEM images of untreated and 40 wt.% HEMA–NIPAM treated bamboo. In the SEM images of the untreated bamboo sample, the pits between the vessels and parenchymal cells are readily visible at both transverse and longitudinal sections. Contrarily, most of the pits in the treated sample were sealed by PHN polymer.

Figure 7A shows the binarized images of untreated and 40 wt.% HEMA–NIPAM solution treated bamboo on cross-sections. The cell wall thickness of twelve groups in each treatment was measured and their mean values are shown in Figure 7B. The average cell wall thickness of untreated and HEMA–NIPAM treated bamboo is 2.48 µm and 4.00 µm, respectively. The cell wall thickness of the treated wood increase by 61.3%, which is possibly related to the swelling of PHN polymers in the micro-voids of the cell wall. This observation is further confirmed by the significant increases in both volume percent gain (VPG) and weight percent gain (WPG), as shown in Figure 7C.

The control group was impregnated with DI water and its VPG and WPG are 1.1% and −1.6%, respectively. The negative WPG is possibly related to the leaching of extractives during the water impregnation process. Statistically, the VPGs of all treated bamboo are not significantly different but much higher than that of the control group. For WPG, there are no significant differences among the 10 wt.% solution treated bamboo. In contrast, the WPG of 20 wt.% and 40 wt.% HEMA–NIPAM treated bamboo is 3.9% and 18.5%, respectively, which is significantly higher than the rest treatments. The increases in cell wall thickness, VPG and WPG of 40 wt.% HEMA–NIPAM treated bamboo, as compared to control, indicate the penetration of PHN in the bamboo cell wall.

### 3.4. Dimensional Stability of PHN Polymer Treated Bamboo

The dimensional stability of bamboo is one of the most important parameters for its applications. Figure 8A shows the macrostructure of specimens after three wetting–drying cycles and soaking–drying cycles treatment. Comparing with the control, no significant cracking appeared on the treated bamboo after three cycles of soaking–drying test. As shown in Figure 8B, all ASE values of treated samples are positive after three-cycle tests, indicating that the treated bamboo presented better dimensional stability than control. Both 10 wt.% NIPAM and 10 wt.% HEMA treated bamboo showed ASE of around 25%, which is significantly lower than those treated with HN polymers due to the high hydrophilicity of PNIPAM and PHEMA. ASE of PHN treated bamboo increases as a function of HN concentrations, with a maximum ASE of 49.4% and 41.7%, respectively, after the wetting–drying cycle and soaking–drying cycle, at a concentration of 40 wt.%. The improved dimensional stability of PHN-treated bamboo indicates that the polymer restrained the cells from deformation and possibly blocked water molecule movement pathways inside the bamboo cell wall.

### 3.5. Mold Resistance of PHN Polymer Treated Bamboo

The average infected area of untreated and treated bamboo after 30 days of mold resistance test is presented in Figure 9. The untreated group was completely infected on day 16. In contrast, the mold infection rates of all the treated bamboo were dramatically suppressed due to the intrinsic cytophobic properties of PHEMA and PNIPAM [27,28]. In particular, NIPAM treated samples demonstrate a higher anti-mold efficacy than that treated with PHEMA alone. The anti-mold properties of PHN treated bamboo were also significantly improved with an average final infected area of 13.5% by day 30 at a treating concentration of 40 wt.%, which is significantly lower than previously reported studies with infection rates over 80% [29,30]. The improved anti-mold resistance of PHN treated bamboo could be attributed to both PHN’s inherent antimicrobial characteristics and its penetration in bamboo, which act as toxic food sources for mold as well as block the cell wall micropores, thus leading to the reduction of diffusion of water and other molecules into the cell wall.

## 4. Conclusions

In this study, PHN polymer was introduced in bamboo cells by in situ polymerization, which has significantly improved the dimension stability and mold resistance of the treated bamboo. Specifically, PHN with a low swelling capacity was obtained through optimizing the synthesis conditions of the polymer by single-factor experiment design. FTIR, TGA and SEM analysis confirmed the formation of PNH, which has a relatively low swelling capacity, high thermal stability and dense spatial structure, respectively. SEM images from PHN treated bamboo shows the presence of PNH polymer in both the cell wall and cell lumen. PHN treated bamboo has significantly improved dimensional stability, as compared to control, with anti-swelling efficiency of 49.4% and 41.7%, respectively, after wetting–drying cycles and soaking–drying cycles. The mold resistance of PHN treated bamboo was also dramatically enhanced with an area infection of 13.5% while the control group was totally covered by the mold after a 30-day mold resistance test. The approach outlined in this study demonstrated a promising method of producing exterior used bamboo that has both high dimensional stability and excellent mold resistance using PHN polymer.

## Figures and Tables

**Figure 1 polymers-12-01584-f001:**
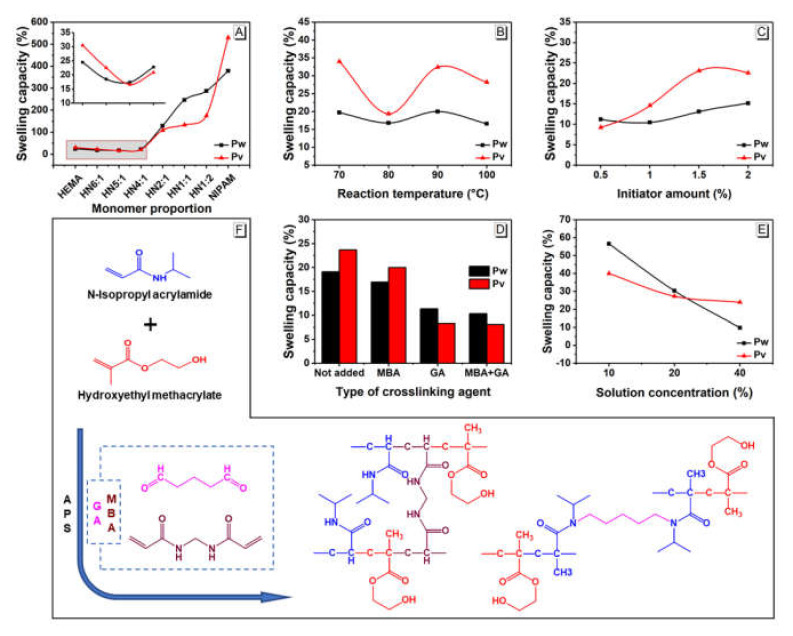
The influence of different factors on swelling capacities of PHN: (**A**) monomer molar ratio, (**B**) reaction temperature, (**C**) initiator amount, (**D**) crosslinking agent type, (**E**) solution concentration, and (**F**) possible reactions between the monomers and crosslinking agents.

**Figure 2 polymers-12-01584-f002:**
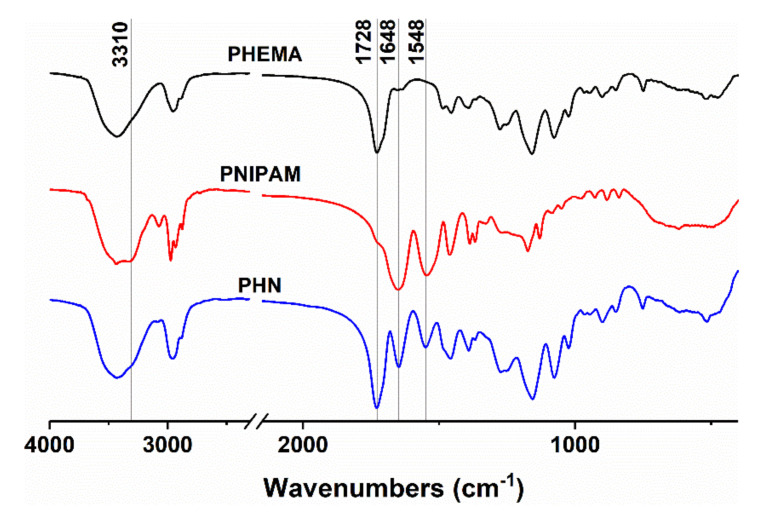
FT-IR spectrum of Poly-hydroxyethyl methacrylate (PHEMA), Poly-N-isopropyl acrylamide (PNIPAM) and poly-HEMA and NIPAM (PHN).

**Figure 3 polymers-12-01584-f003:**
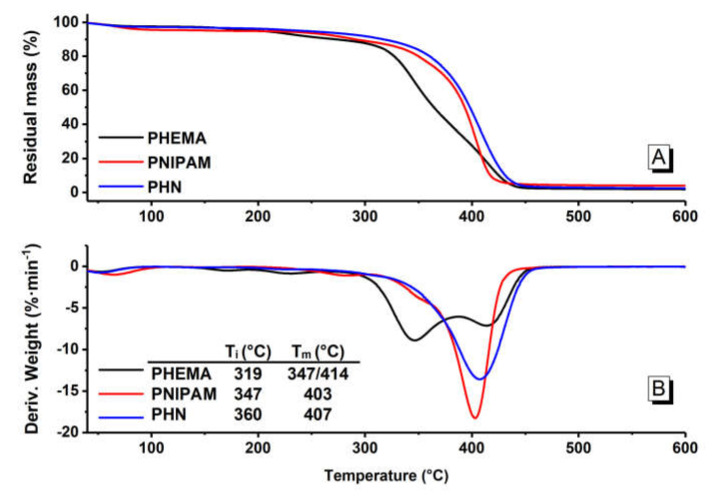
The thermogravimetric analyzer (TG) (**A**) and derivative thermogravimetry (DTG) (**B**) profiles of PHEMA, PNIPAM, and PHN.

**Figure 4 polymers-12-01584-f004:**
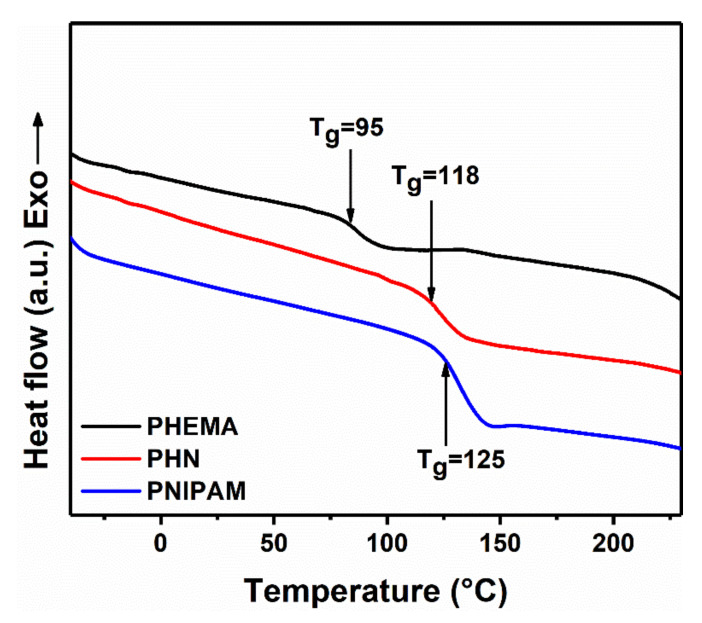
The differential scanning calorimetry (DSC) thermograms of PHEMA, PHN, and PNIPAM (The second run of all the DSC curves was presented here).

**Figure 5 polymers-12-01584-f005:**
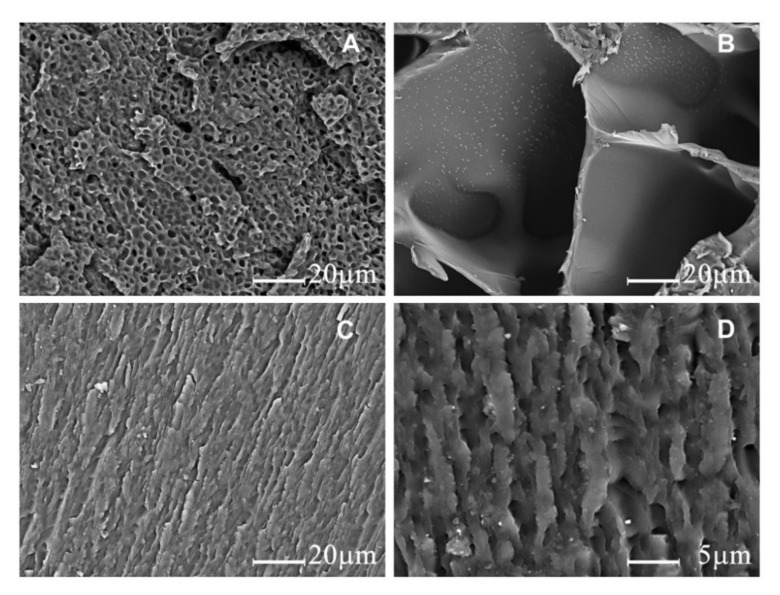
SEM micrographs of polymers (**A**) PHEMA, (**B**) PNIPAM, (**C**) PHN at a magnification of 1000×, and (**D**) PHN at a magnification of 3000×.

**Figure 6 polymers-12-01584-f006:**
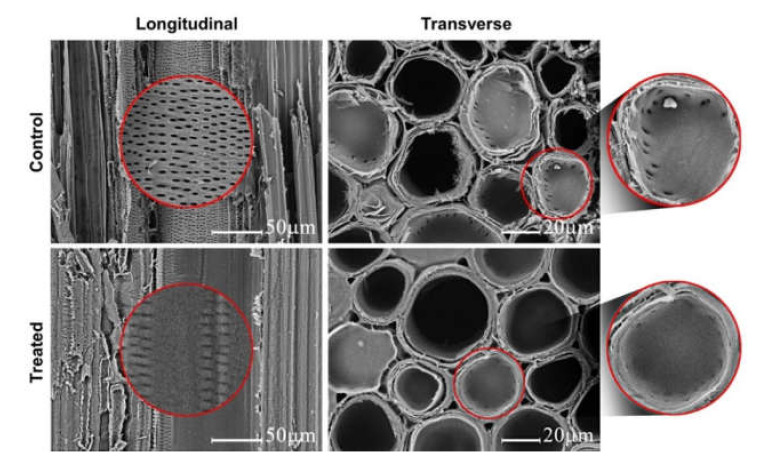
SEM micrographs of control and 40 wt.% HEMA–NIPAM treated bamboo.

**Figure 7 polymers-12-01584-f007:**
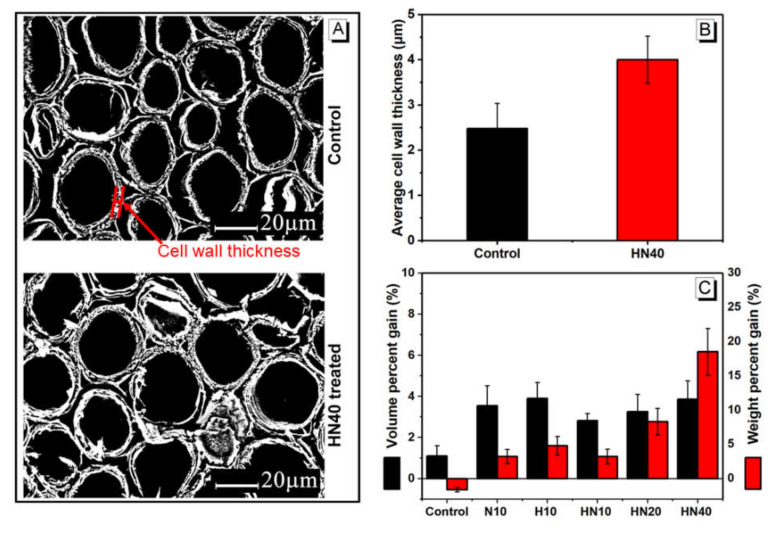
PHN polymer in bamboo cells: (**A**) binarized images, (**B**) average cell wall thickness of control and 40 wt.% HEMA–NIPAM treated bamboo, and (**C**) volume percent gain (VPG) and weight percent gain (WPG) of control and treated bamboo at various treatment conditions.

**Figure 8 polymers-12-01584-f008:**
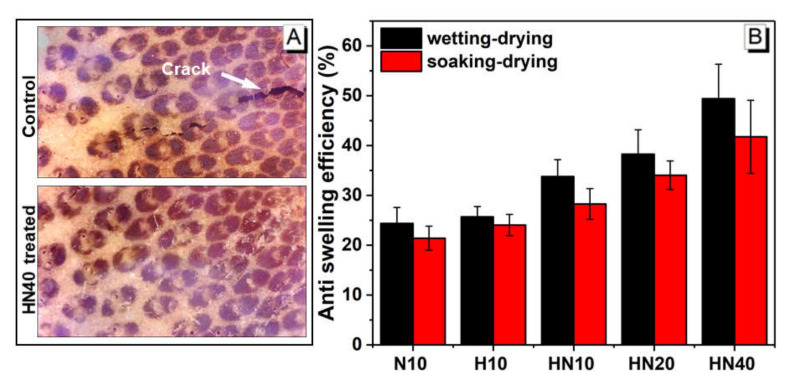
(**A**) The macrostructure of control and treated bamboo (cross-section) after soaking–drying tests (**B**) The anti-swelling efficiency (ASE) of treated bamboo in a wetting–drying cycle and soaking–drying cycle.

**Figure 9 polymers-12-01584-f009:**
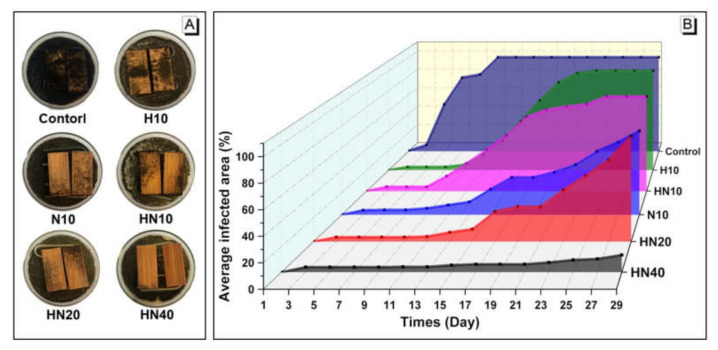
(**A**) Photographs of bamboo with different treatment after 30-day mold exposure and (**B**) mold average infected area of bamboo samples over the 30-day testing period.

**Table 1 polymers-12-01584-t001:** Experiment design of poly-hydroxyethyl methacrylate (HEMA) and *N*-isopropyl acrylamide (NIPAM) (PHN) synthesis using different molar ratios of HEMA and NIPAM.

Reaction Condition	Codes							
	HEMA	HN61	HN51	HN41	HN21	HN11	HN12	NIPAM
HEMA: NIPAM (molar ratio)	1:0	6:1	5:1	4:1	2:1	1:1	1:2	0:1
Concentration (wt.%)	40							15
others	APS = 1 wt.%, GA = 1 wt.%, MBA = 1 wt.%, T = 80 °C, t = 2 h,							

**Table 2 polymers-12-01584-t002:** Different polymer concentrations for bamboo treatment.

Codes	Modifiers	Concentrations (wt.%)
N10	NIPAM	10
H10	HEMA	10
HN10	HEMA–NIPAM	10
HN20	HEMA–NIPAM	20
HN40	HEMA–NIPAM	40

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
