# Peer review of "Improved Dimensional Stability and Mold Resistance of Bamboo via In Situ Growth of Poly(Hydroxyethyl Methacrylate-N-Isopropyl Acrylamide)"

_polymers, 2020, doi:10.3390/polym12071584_

Round 1

Reviewer 1 Report

In this study, the conditions to synthesize PHN polymer with the lowest swelling capacity was optimized and then evaluate the chemical, physical and morphological properties of the treated bamboo with the optimized formulations. The distribution of PHN in bamboo was visualized using SEM and quantified by volume percentage gain and weight percentage gain. The dimensional stability and mold resistance of the treated bamboo were also evaluated. The manuscript is well written, contain new information and novelty. The results interpreted correctly and the mechanism behind well described. I warm-heartedly recommend this manuscript for publication as it is, because of its complex scientific merit and correct presentation. 

Reviewer 2 Report

The authors present a throughout experimental study on in situ polymerisation of Bamboo. The work is clearly application oriented to reduce mold for outdoor use of bamboo as construction material. 

The results are of interest for industrial application also for furnitures made of bamboo. 

While the study investigates dimensional studies over 30 days, mechanical stability is not addressed. Also the authors do unfortunately not discuss environmental impact of the used materials. 

From the scientific point of view the study is however self-consistent, well performed and documented and of interest for the scientific community. 

I thus recommend publication of the manuscript.